# The Comparison of Three Predictive Indexes to Discriminate Malignant Ovarian Tumors from Benign Ovarian Endometrioma: The Characteristics and Efficacy

**DOI:** 10.3390/diagnostics12051212

**Published:** 2022-05-12

**Authors:** Shoichiro Yamanaka, Naoki Kawahara, Ryuji Kawaguchi, Keita Waki, Tomoka Maehana, Yosuke Fukui, Ryuta Miyake, Yuki Yamada, Hiroshi Kobayashi, Fuminori Kimura

**Affiliations:** Department of Obstetrics and Gynecology, Nara Medical University, 840 Shijo-cho, Kashihara 634-8522, Japan; shoichiroyamanaka@naramed-u.ac.jp (S.Y.); kawaryu@naramed-u.ac.jp (R.K.); k178719@naramed-u.ac.jp (K.W.); tmaehana@naramed-u.ac.jp (T.M.); f.yosuke1002@naramed-u.ac.jp (Y.F.); ryuta-miyake@naramed-u.ac.jp (R.M.); yuki0528@naramed-u.ac.jp (Y.Y.); hirokoba@naramed-u.ac.jp (H.K.); kimurafu@naramed-u.ac.jp (F.K.)

**Keywords:** ovarian endometrioma, endometriosis associated ovarian cancer, malignant ovarian tumor, borderline ovarian tumor, CPH index, ROMA index, R2 predictive index

## Abstract

This study aimed to evaluate the prediction efficacy of malignant transformation of ovarian endometrioma (OE) using the Copenhagen Index (CPH-I), the risk of ovarian malignancy algorithm (ROMA), and the R2 predictive index. This retrospective study was conducted at the Department of Gynecology, Nara Medical University Hospital, from January 2008 to July 2021. A total of 171 patients were included in the study. In the current study, cases were divided into three cohorts: pre-menopausal, post-menopausal, and a combined cohort. Patients with benign ovarian tumor mainly received laparoscopic surgery, and patients with suspected malignant tumors underwent laparotomy. Information from a review chart of the patients’ medical records was collected. In the combined cohort, a multivariate analysis confirmed that the ROMA index, the R2 predictive index, and tumor laterality were extracted as independent factors for predicting malignant tumors (hazard ratio (HR): 222.14, 95% confidence interval (CI): 22.27–2215.50, *p* < 0.001; HR: 9.80, 95% CI: 2.90–33.13, *p* < 0.001; HR: 0.15, 95% CI: 0.03–0.75, *p* = 0.021, respectively). In the pre-menopausal cohort, a multivariate analysis confirmed that the CPH index and the R2 predictive index were extracted as independent factors for predicting malignant tumors (HR: 6.45, 95% CI: 1.47–28.22, *p* = 0.013; HR: 31.19, 95% CI: 8.48–114.74, *p* < 0.001, respectively). Moreover, the R2 predictive index was only extracted as an independent factor for predicting borderline tumors (HR: 45.00, 95% CI: 7.43–272.52, *p* < 0.001) in the combined cohort. In pre-menopausal cases or borderline cases, the R2 predictive index is useful; while, in post-menopausal cases, the ROMA index is better than the other indexes.

## 1. Introduction

Ovarian cancer is the fifth leading cause of cancer-related death in women [1]. This disease cannot be diagnosed in the early stages and is called the silent killer [2,3,4]. As such, most ovarian cancer cases are diagnosed at advanced stages [5,6,7], and over 185,000 deaths due to this disease are reported annually worldwide [8,9].

Molecular genetics and morphologic characteristics revealed that ovarian cancer can be divided into two categories, designated types 1 and 2 [10,11,12]. Type 1 tumors show a stepwise progression (adenoma–carcinoma sequence), which comprise endometriosis-associated ovarian cancer (EAOC), such as clear cell carcinoma and low-grade endometrioid carcinoma, as well as mucinous carcinoma and low-grade serous carcinoma [13,14]. Type 2 tumors range from the normal epithelium to precursor lesions, and finally to high-grade serous and endometrioid carcinoma, malignant mixed mesodermal tumors (carcinosarcomas), and undifferentiated carcinoma [13,15]. The former shows low progression but is resistant to chemotherapy; in contrast, the latter is highly progressive but shows vulnerability to chemotherapy [16]. In type 1 ovarian cancer, most EAOC arises from ovarian endometriosis [17,18,19], and there is a major challenge for physicians in the case of early detection/surgical treatment and effects on fertility.

Ovarian endometriosis is defined as the presence of endometrial glands and stroma outside of the uterus, and it is most often detected in the pelvic peritoneum and ovaries [20]. Repeated hemorrhages in the peritoneum or ovaries may contribute to the symptoms of dysmenorrhea [21,22], chronic pelvic pain [23,24], and infertility [25,26], which negatively affect the patients’ quality of life. There is also evidence of an epidemiologic link between iron overload and the various types of human carcinoma, including malignant mesothelioma, renal cell carcinoma, hepatocellular carcinoma, and EAOC [27,28,29,30,31]. We showed that total iron levels of cyst fluid can discriminate EAOC from ovarian endometrioma (OE), with a cutoff point of 64.8 mg/L (sensitivity, 85%; specificity, 98%) [32]; and magnetic resonance (MR) relaxometry, which can noninvasively measure cyst fluid iron concentration, can discriminate with a cutoff point of 12.1 (sensitivity, 86%; specificity, 94%) [33,34]. Moreover, we showed a novel predictive tool in the R2 predictive index, which requires tumor diameter (mm) and blood tumor marker as CEA (ng/mL). This index is useful and valuable for the detection of the malignant transformation of endometrioma (i.e., EAOC), with good accuracy (sensitivity, 82%; specificity, 68%) [35]. In clinical practice, ultrasound is the most powerful tool to detect ovarian tumors and can differentiate between OE and malignant ovarian tumors (i.e., IOTA classification) [36,37]; however, a good level of understanding and training are needed to score the system. There are some effective tools to discriminate malignant ovarian tumors from benign tumors [35,36,37,38,39,40,41,42,43]. The risk of ovarian malignancy algorithm (ROMA) index value is an algorithm that takes into account the levels of carbohydrate antigen125 (CA125) and human epididymis protein 4 (HE4), together with menopausal status, using quantitative and objective parameters; and the Copenhagen (CPH) index takes into account HE4, CA125, and age, rather than menopausal status, with different definitions.

The current study aimed to compare the efficacy of these predictive tools and investigate the characteristics of these indexes.

## 2. Materials and Methods

### 2.1. Patients

A list of patients with primary, previously untreated, histologically-confirmed ovarian tumors who were treated at Nara Medical University Hospital between January 2008 and July 2021 was generated from our institutional registry. We retrospectively included in this study the following cases of OE as benign ovarian tumor and EAOC cases as malignant tumor with available blood samples for tumor marker calculations. All of the OE and EAOC cases were histologically confirmed. Written consent for the use of the patients’ clinical data for research was obtained at the first hospitalization, and after approval by the Ethics Review Committee of the Nara Medical Hospital; the opt-out form was provided through our institutional homepage. The current study consisted of three cohorts: the pre-menopausal, post-menopausal, and combined cohorts. Pre-menopause and post-menopause were divided by age, namely under 50 years old was defined as pre-menopause and over 50 years old as post-menopause. The pre-menopausal cohort included 115 patients with newly diagnosed ovarian tumors. A total of 56 patients were included in the post-menopause cohort. No patients had undergone chemotherapy or radiotherapy for the ovarian tumors prior to treatment. Patients with OE mainly received laparoscopic surgery, and the patients suspected of harboring malignant tumors underwent laparotomy. The following factors were collected through a chart review of the patients’ medical records: age; body mass index (BMI); parity; postoperative diagnosis, including FIGO (The International Federation of Gynecology and Obstetrics) stage; the date of surgery; tumor diameter; menopausal status; and pre-treatment blood test results, including CA125, carbohydrate antigen 19-9 (CA 19-9), carcinoembryonic antigen (CEA), and HE4 as a tumor marker. The cases shared with a previous study [35] were 72 cases (42.1%).

### 2.2. Tumor Imaging and Diagnoses

All patients first visited the outpatient clinic and underwent internal examination, including ultrasound followed by routine MR imaging using T1W and T2W sequences. Tumor diameter was recorded as the largest diameter among axial, sagittal, and coronal imaging. Patients were largely diagnosed with OE or EAOC by MRI, and this was confirmed by the histological examination using the surgically removed tumors by at least two pathologists who were blinded to the study. The number of EAOC cases that were histologically proven as arising from endometriosis were 41 cases (54.7%).

### 2.3. Detection of CA125, CA19-9, CEA, and HE4 Concentrations

Samples were collected from all the patients prior to surgery using blood collection tubes without anticoagulants. Each blood sample was centrifuged at 3000 rpm and stored at −80 °C until use. Tumor markers including CA125 (ARCHITECT CA125 II, Abbott Japan LLC, Tokyo, Japan), CA19-9 (CL AIA-PACK^®^ SLa, Tosoh Corporation, Tokyo, Japan), CEA (CL AIA-PACK^®^ CEA, Tosoh Corporation, Tokyo, Japan), and HE4 (ARCHITECT HE4, Abbott Japan LLC, Tokyo, Japan) were measured using a chemiluminescence immunoassay, according to the manufacturer’s instructions. Serum samples in dry ice were transported to the Tosoh diagnostics product divisions (Tosoh Corporation, Kanagawa, Japan), and CA19-9 and CEA concentrations were determined immediately. HE4 and CA125 (ARCHITECT CA125 II) were measured at BML INC., Tokyo, Japan. In case of CA125 and HE4 levels under the limit, we recorded the lower limit of calibration as 1 (U/mL) and 20 (pmol/L), respectively. Measurements were performed by clinical laboratory technologists who were blinded to the study.

### 2.4. Calculation of the ROMA, the CPH, and the R2 Predictive Value

Using the concentrations of CA125, HE4, and CEA, we calculated the Copenhagen (CPH) index, the risk of ovarian malignancy algorithm (ROMA) index, and the R2 predictive index, according to the mathematical equations presented below.

The ROMA index was calculated using the following equations [44]:Pre-menopausal predictive index (PI) = –12.0 + 2.38 × LN(HE4) + 0.0626 × LN(CA 125)Post-menopausal PI = −8.09 + 1.04 × LN(HE4) + 0.732 × LN(CA125)ROMA(%) = exp(PI)/[1 + exp(PI)] × 100(1)
LN = natural log function and exp(PI) = e^PI^.

The CPH index was calculated using the following equations [45]:PI = −14.0647 + 1.0649 × log_2_(HE4) + 0.6050 × log_2_(CA125) + 0.2672 × (age/10)
CPH-I = exp(PI)/[1 + exp(PI)] × 100(2)
The R2 predictive index was calculated using the following equations [35]:


[R2 predictive index] = 27.27 − 7.90 × 10^−2^ × (Tumor diameter) − 1.31 × (CEA)(3)


### 2.5. Statistical Analysis

Analyses were performed using SPSS version 25.0 (IBM SPSS, Armonk, NY, USA). The differences of each factor, including the CPH index, the ROMA index, and the R2 predictive index among groups, were compared using a Mann–Whitney U test or Kruskal–Wallis one-way ANOVA test. The receiver operating characteristic (ROC) curve analysis was performed to determine the cut-off value for predicting malignant ovarian tumors in each pre-menopausal, post-menopausal, and combined (pre- and post-menopause) cohort. The cut-off value was based on the highest Youden index (i.e., sensitivity + specificity − 1). We next used a logistic regression analysis to assess the risk factors for malignant ovarian tumors (i.e., EAOC). A two-sided *p* < 0.05 was considered as indicating a statistically significant difference.

## 3. Results

### 3.1. Patients

From January 2008 to July 2021, a total of 171 patients included in this study were divided as follows: 115 patients who were under 50 years old as the pre-menopausal cohort, and 56 patients over 50 years old as the post-menopausal cohort. The combined cohort consisted of the pre- and post-menopausal cohorts. The demographic and clinical characteristics of the combined cohort are outlined in Table 1. In the combined cohort, a post-operative diagnosis of OE was found in 96 (56.1%) and malignant tumors in 75 (43.9%) patients, including eight cases of borderline tumor. In this cohort, there was significant differentiation in age, BMI, gravida, parity, cyst size, menopausal status, and tumor laterality. Table 2 shows the distribution of each biological marker. CEA, HE4, CA125, and D-dimer reached significant differentiation between a benign tumor and malignant tumor.

### 3.2. The Characteristics of Each Biological Marker in Each Cohort

The results of the ROC curve analysis based on the detection of malignant tumors are shown in Figure 1, concerning each predictive index, and in Figure 2 and Figure 3 regarding other biological markers. The optimal cutoff value was determined by analyzing the ROC curve among malignant ovarian tumors and OE. Table 3 shows the cut-off values discriminating benign from malignant tumors for each cohort. In the post-menopause cohort, CEA and tumor diameter, which comprise the R2 predictive index, did not reach significant differentiation; on the other hand, CA125, comprising the CPH index and the ROMA index, in the pre-menopause cohort did not reach significant differentiation. This characteristic influences the AUC of each index, including the CPH index, the ROMA index, and the R2 predictive index.

### 3.3. The Usefulness of Each Index in Discriminating OE and Malignant Ovarian Tumors

In the combined cohort, some factors indicating malignant ovarian tumors (i.e., EAOC) were extracted using a univariate analysis (Table 4). A multivariate analysis confirmed that the ROMA index, the R2 predictive index, and tumor laterality were extracted as independent factors for predicting malignant tumors (HR: 222.14, 95% confidence interval (CI): 22.27–2215.50, *p* < 0.001; HR: 9.80, 95% CI: 2.90–33.13, *p* < 0.001; HR: 0.15, 95% CI: 0.03–0.75, *p* = 0.021, respectively). Furthermore, excluding the CPH index, the ROMA index, and the R2 predictive index, a multivariate analysis showed that laterality, tumor diameter, D-dimer, CEA, and HE4 were the independent factors (HR: 0.22, 95% CI: 0.08–0.65, *p* = 0.006; HR: 12.68, 95% CI: 4.21–38.22, *p* < 0.001; HR: 5.13, 95% CI: 1.81–14.53, *p* = 0.002; HR: 4.36, 95% CI: 1.75–10.85, *p* = 0.002; HR: 3.85, 95% CI: 1.37–10.82, *p* = 0.011, respectively) (Table 4). In the pre-menopausal cohort, a multivariate analysis confirmed that the CPH index and the R2 predictive index were extracted as independent factors for predicting malignant tumors (HR: 6.45, 95% CI: 1.47–28.22, *p* = 0.013; HR: 31.19, 95% CI: 8.48–114.74, *p* < 0.001, respectively). Excluding the CPH index, the ROMA index, and the R2 predictive index, a multivariate analysis showed that laterality, tumor diameter, and HE4 were the independent factors (HR: 0.15, 95% CI: 0.02–0.81, *p* = 0.028; HR: 11.78, 95% CI: 3.09–44.93, *p* < 0.001; HR: 47.94, 95% CI: 4.01–572.03, *p* = 0.002, respectively) (Table 5). In the combined cohort, the ROMA index showed the highest diagnostic accuracy (Table 6) and a similar result as the univariate analysis (Table 4). However, in the pre-menopausal cohort, the ROMA index showed the highest accuracy (Table 6), but this did not remain using a univariate analysis (Table 5).

### 3.4. The Usefulness of the R2 Predictive Index in Discriminating OE from Borderline Tumors

In the combined cohort, some factors indicating a borderline tumor were extracted by the univariate analysis (Table 7). Multivariate analysis confirmed that the R2 predictive index was only extracted as an independent factor for predicting malignant tumors (HR: 45.00, 95% CI: 7.43–272.52, *p* < 0.001). When excluding the CPH index, the ROMA index, and the R2 predictive index from the factor and including tumor diameter, CEA, HE4, and CA125, only tumor diameter was indicated as an independent factor (HR: 7.33, 95% CI: 1.32–40.48, *p* = 0.022) (Table 7).

### 3.5. The Differentiation of R2 Predictive Value between OE and Borderline Tumor or Advanced Malignant Tumors

In the combined cohort, the R2 predictive index, the ROMA index, and the CPH index showed significant differentiation among ovarian endometriosis, borderline tumor, and carcinoma (Figure 4). The ROMA index and the CPH index could discriminate the carcinoma from the others; on the contrary, the R2 predictive index discriminated the endometriosis from malignant tumors (Figure 4, Table 8).

## 4. Discussion

In the current study, the ROMA index, the CPH index, and the R2 predictive index were shown to be effective tools to discriminate benign OE from EAOC, and showed similar results to those reported previously [46,47]. In particular, in the combined cohort, the ROMA index was the most effective predictor among the three indexes (Table 4); however, in the pre-menopausal cohort, the R2 predictive index was more effective than the others (Table 5) for discriminating malignant tumors. This is partly because HE4 and CA125, which consist of the CPH and the ROMA index have a weaker ability to discriminate malignancy in pre-menopausal durations; on the other hand, CEA, which consists of the R2 predictive value, was stronger in the pre-menopausal cohort than HE4 and CA125 (Table 3). Serum CA125 levels are frequently measured when ovarian cysts are observed, in order to rule out a malignant tumor. However, it is well known that elevated serum CA125 levels are not only seen in endometrioma [48], but also in adenomyosis [49] or menstrual cycle [50], thus giving a high rate of false positives [51,52]. This was confirmed in a recent Cochrane review, which reported that among the 97 biomarkers studied, CA125 was the only marker that is elevated in cases of endometrioma, with 40% sensitivity and 91% specificity, with a cut-off limit of 35 U/mL [53].

On the other hand, HE4 is the most promising. HE4 protein is encoded by the WAP four-disulfide core domain 2 (WFDC2) [54], which was found to be highly expressed in ovarian carcinoma, especially in serous and endometrioid cancers [55,56]. Unlike CA125, HE4 is not overexpressed in benign ovarian disease, normal ovarian tissue, or tumors with low malignant potential [55]. Terlikowska KM et al. reported that the HE4 level in serum elevates with age, and the specificity was better in post-menopausal patients than in pre-menopausal patients [57]. This trend is similar to that in our results, in which the CPH and the ROMA index were useful tools to discriminate malignancy in post-menopausal patients. In particular, in pre-menopausal patients there is a major challenge in choosing the surgical method (i.e., laparotomy or laparoscopic surgery), and this index could be helpful for the physician.

We previously reported that OE has a higher iron concentration than EAOC and can discriminate either cyst fluid iron concentration or transverse magnetic relaxation rate R2 or R2* value, using a complex, chemical shift-encoded MR examination [37,38]. However, no evidence concerning the standpoint of borderline tumor (i.e., the degree of iron concentration or R2 value) exists, because of the rare incidence of this disease. We demonstrated that the R2 predictive index was the independent factor to discriminate borderline tumor from OE in the combined cohort (Table 7). Moreover, the R2 predictive index of OE was higher than other malignant tumors with significant differentiation (Table 8). We can hypothesize that borderline tumors could show lower iron concentrations than OE, and this may discriminate benign OE from EAOC, even in borderline cases, by iron concentration and transverse magnetic relaxation rate R2 or R2* value.

This study has some limitations. The first limitation is that the number of OE in post-menopausal patients was too small to assess the effectiveness of these indexes in the post-menopausal cohort. Second, the sample sizes of the borderline ovarian tumor and phenotype were too small to conclude the efficacy of the R2 predictive index in discriminating borderline tumors from endometriosis, and further case accumulation is needed.

## 5. Conclusions

In conclusion, in pre-menopausal cases or borderline cases, the R2 predictive index is useful; and in post-menopausal cases, the ROMA index is better than the other indexes.

## Figures and Tables

**Figure 1 diagnostics-12-01212-f001:**
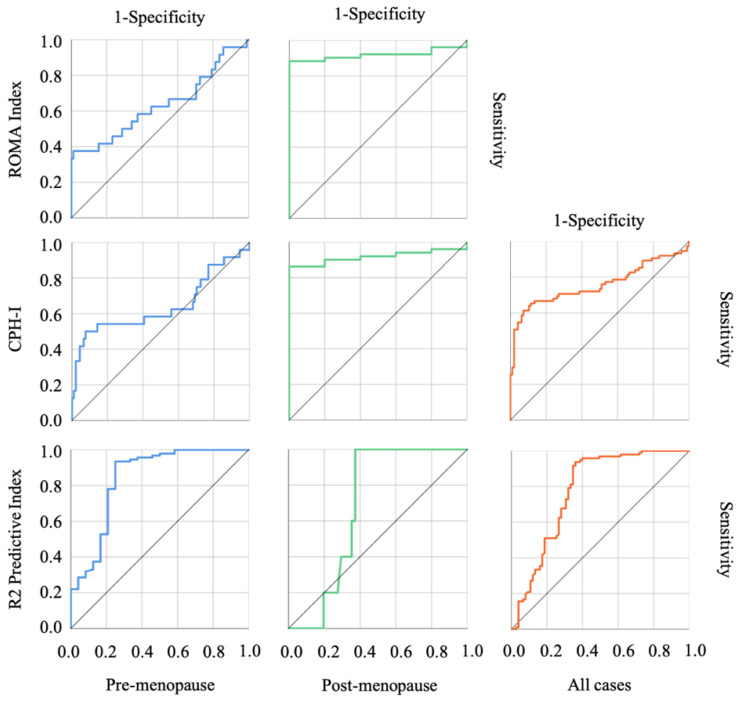
The ROC curves of each predictive index in the combined cohort. The row indicates each predictive index and the column indicates each cohort. The R2 predictive index showed a high AUC in the pre-menopausal cohort; on the contrary, the ROMA and CPH indexes showed high AUCs in post-menopausal cohort.

**Figure 2 diagnostics-12-01212-f002:**
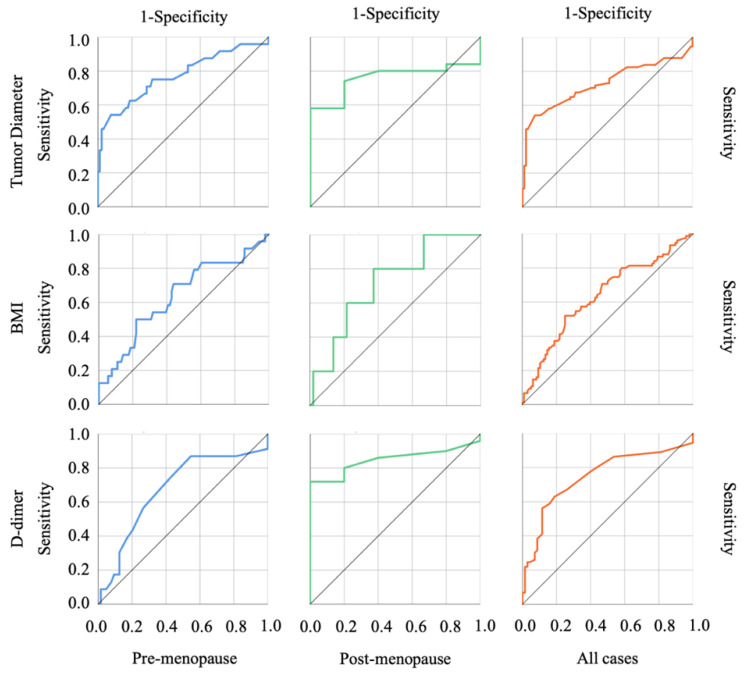
The ROC curves of other factors. The row indicates each factor, and the column indicates each cohort.

**Figure 3 diagnostics-12-01212-f003:**
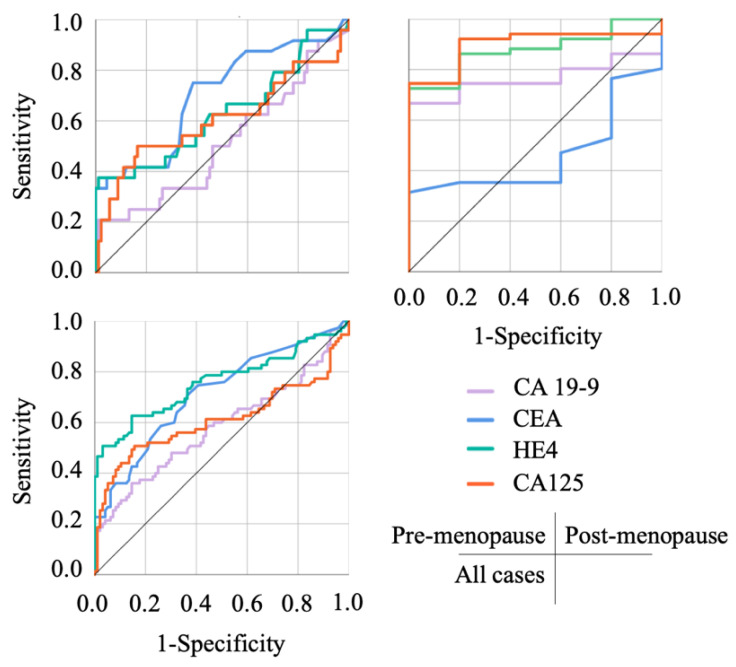
The ROC curves of each tumor marker. CEA showed a higher AUC than HE4 and CA125 in the pre-menopausal cohort; however, in the post-menopausal cohort HE4 and CA125 increased their AUC in the post-menopausal cohort.

**Figure 4 diagnostics-12-01212-f004:**
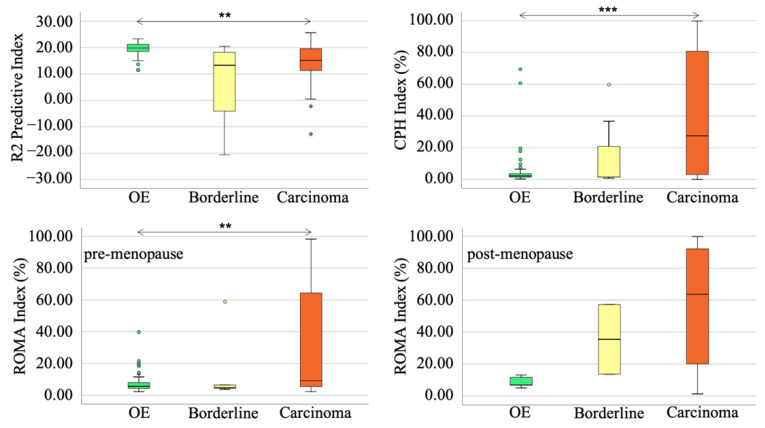
To discriminate borderline tumors from ovarian endometriosis, the R2 predictive index could be the most effective tool. ** *p* < 0.01 vs. others., *** *p* < 0.001 vs. others. The circles represent outliers. There were only two borderline cases in post-menopausal cohort (lower right).

**Table 1 diagnostics-12-01212-t001:** Demographic and clinical characteristics of the combined cohort.

	Benign Tumor (OE)	Malignant Tumor (EAOC)	*p*-Value
Number	*n* = 96	*n* = 75	
Age (years)			
Median (range)	37.00 (18–63)	54.00 (21–82)	
Mean ± SD	36.40 ± 8.82	54.36 ± 11.63	<0.001
BMI			
Median (range)	20.05 (14.52–34.25)	21.98 (15.20–36.00)	
Mean ± SD	20.75 ± 3.55	22.49 ± 4.22	0.002
Gravida			
0	55	25	
≥1	41	50	0.001
Parity			
0	59	26	
≥1	37	49	<0.001
FIGO sage	–	I (*n* = 49), II (*n* = 3), III (*n* = 15), IV (*n* = 8)	
Subtype	Endometrioma (*n* = 96)	Endometrioid carcinoma (*n* = 27)	
		CCC (*n* = 40)	
		SMBT (*n* = 8)	
Cyst size (mm)			
Median (range)	64.50 (38.00–185.00)	105.00 (16.50–350.00)	
Mean ± SD	67.79 ± 22.97	110.05 ± 60.85	<0.001
Menopause			
Yes	5	51	
No	91	24	<0.001
Laterality		*	
Unilateral	56	60	
Bilateral	40	14	0.001

OE ovarian endometrioma, EAOC endometriosis-associated ovarian cancer, BMI body mass index, FIGO The International Federation of Gynecology and Obstetrics, CCC clear cell carcinoma, SMBT seromucinous borderline tumor. * missing data.

**Table 2 diagnostics-12-01212-t002:** Tumor markers in blood samples in the combined cohort.

	Benign Tumor (OE)	Malignant Tumor (EAOC)	*p*-Value
Number	*n* = 96	*n* = 75	
CA 19-9 (U/mL)			
Median (range)	23.30 (0.50–1085.70)	29.50 (0.00–8953.10)	
Mean ± SD	48.14 ± 118.09	391.93 ± 1305.99	0.068
CEA (ng/mL)			
Median (range)	1.50 (0.60–5.20)	2.20 (0.70–30.00)	
Mean ± SD	1.75 ± 0.99	4.05 ± 5.25	<0.001
HE4 (pmol/L)			
Median (range)	42.30 (28.10–107.70)	72.7 (28.7–1873.70)	
Mean ± SD	45.19 ± 12.26	215.52 ± 336.46	<0.001
CA125 (U/mL)			
Median (range)	58.25 (10.10–5525.20)	147.20 (1.00–9426.00)	
Mean ± SD	159.70 ± 575.42	691.53 ± 1402.11	0.013
Hb (g/mL)			
Median (range)	12.60 (8.90–14.60)	12.80 (4.60–15.70)	
Mean ± SD	12.60 ± 1.06	12.46 ± 1.88	0.691
D-dimer (µg/mL)			
Median (range)	0.70 (0.50–8.40)	1.30 (0.40–34.70)	
Mean ± SD	0.99 ± 1.11	3.05 ± 4.87	<0.001

OE ovarian endometrioma, EAOC endometriosis-associated ovarian cancer, CA 19-9 carbohydrate antigen 19-9, CEA carcinoembryonic antigen, HE4 human epididymis protein 4, CA125 carbohydrate antigen125, Hb hemoglobin.

**Table 3 diagnostics-12-01212-t003:** The cut-off values discriminating EAOC from benign OE in the pre-, post-menopausal, and combined cohorts.

	AUC	*p*-Value	Cut-Off Value	Sensitivity	Specificity	PPV	NPV
CA 19-9 (U/mL)							
Pre-menopause	0.511	0.872	–	–	–	–	–
Post-menopause	0.765	0.062	–	–	–	–	–
Combined	0.581	0.068	–	–	–	–	–
CEA (ng/mL)							
Pre-menopause	0.704	0.002	1.55	0.750	0.615	33.96	90.32
Post-menopause	0.465	0.796	–	–	–	–	–
Combined	0.714	<0.001	1.65	0.707	0.635	60.22	73.49
HE4 (pmol/L)							
Pre-menopause	0.631	0.049	82.90	0.375	0.989	90.00	85.71
Post-menopause	0.878	0.006	54.10	0.725	1.000	100.00	26.31
Combined	0.758	<0.001	54.65	0.627	0.854	77.04	74.54
CA125 (U/mL)							
Pre-menopause	0.606	0.112	–	–	–	–	–
Post-menopause	0.898	0.004	15.00	0.922	0.800	97.91	50.00
Combined	0.610	0.013	146.15	0.507	0.844	71.69	68.64
Tumor diameter (mm)							
Pre-menopause	0.772	<0.001	97.50	0.542	0.923	65.00	88.42
Post-menopause	0.758	0.059	–	–	–	–	–
Combined	0.726	<0.001	97.50	0.541	0.927	85.10	71.77
BMI							
Pre-menopause	0.636	0.041	21.94	0.500	0.780	37.50	85.54
Post-menopause	0.718	0.111	–	–	–	–	–
Combined	0.636	0.002	21.94	0.520	0.750	61.90	66.66
D-dimer (µg/mL)							
Pre-menopause	0.675	0.013	0.65	0.870	0.453	32.78	92.59
Post-menopause	0.848	0.011	0.95	0.720	1.000	100.00	26.31
Combined	0.748	<0.001	1.15	0.562	0.884	75.00	71.30
CPH-I (%)							
Pre-menopause	0.642	0.032	6.564	0.500	0.923	63.15	87.50
Post-menopause	0.918	0.002	1.884	0.863	1.000	100.00	41.66
Combined	0.758	<0.001	6.564	0.613	0.927	86.79	75.42
ROMA Index (%)							
Pre-menopause	0.633	0.046	24.78	0.375	0.989	90.00	85.71
Post-menopause	0.918	0.002	13.23	0.882	1.000	100.00	45.45
Combined	–	–	–	–	–	98.18	81.89
R2 Predictive Index							
Pre-menopause	0.840	<0.001	16.95	0.934	0.750	75.00	93.40
Post-menopause	0.684	0.177	18.39	1.000	0.627	100.00	20.83
Combined	0.777	<0.001	16.95	0.938	0.640	88.88	76.92

CA 19-9 carbohydrate antigen 19-9, CEA carcinoembryonic antigen, HE4 human epididymis protein 4, CA125 carbohydrate antigen125, BMI body mass index, CPH-I Copenhagen index, ROMA risk of ovarian malignancy algorithm, PPV positive predictive value, NPV negative predictive value, AUC area under curve.

**Table 4 diagnostics-12-01212-t004:** Univariate and multivariable analysis of the predictive factors of EAOC in the combined cohort.

		Univariate Analysis		Multivariate Analysis			
		Risk Ratio (95% CI)	*p*-Value	Risk Ratio (95% CI)	*p*-Value	Risk Ratio (95% CI)	*p*-Value
CPH-I	≤6.564	1.00 (referent)				—	—
(%)	>6.564	20.16 (8.20–49.54)	<0.001			—	—
ROMA Index		1.00 (referent)		1.00 (referent)		—	—
(%)		244.28 (31.96–1866.91)	<0.001	222.14 (22.27–2215.50)	<0.001	—	—
R2 Predictive	≤16.95	1.00 (referent)		1.00 (referent)		—	—
Index	>16.95	26.66 (10.29–69.05)	<0.001	9.80 (2.90–33.13)	<0.001	—	—
Gravida	0	1.00 (referent)					
	≥1	2.68 (1.43–5.02)	0.002				
Parity	0	1.00 (referent)					
	≥1	3.00 (1.60–5.63)	0.001				
Laterality	Uni-	1.00 (referent)		1.00 (referent)		1.00 (referent)	
	Bi-	0.32 (0.16–0.66)	0.002	0.15 (0.03–0.75)	0.021	0.22 (0.08–0.65)	0.006
BMI	≤21.94	1.00 (referent)					
	>21.94	3.25 (1.70–6.20)	<0.001				
Tumor diameter	<97.50	1.00 (referent)		—	—	1.00 (referent)	
(mm)	≥97.50	14.53 (5.94–35.49)	<0.001	—	—	12.68 (4.21–38.22)	<0.001
D-dimer	<1.15	1.00 (referent)				1.00 (referent)	
(µg/mL)	≥1.15	7.45 (3.60–15.42)	<0.001			5.13 (1.81–14.53)	0.002
CEA	<1.65	1.00 (referent)		—	—	1.00 (referent)	
(ng/mL)	≥1.65	4.19 (2.19–8.02)	<0.001	—	—	4.36 (1.75–10.85)	0.002
HE4	<54.65	1.00 (referent)		—	—	1.00 (referent)	
(pmol/L)	≥54.65	9.83 (4.71–20.50)	<0.001	—	—	3.85 (1.37–10.82)	0.011
CA125	<146.15	1.00 (referent)		—	—		
(U/mL)	≥146.15	5.54 (2.71–11.31)	<0.001	—	—		

CPH-I Copenhagen index, ROMA risk of ovarian malignancy algorithm, BMI body mass index, CEA carcinoembryonic antigen, HE4 human epididymis protein 4, CA125 carbohydrate antigen125.

**Table 5 diagnostics-12-01212-t005:** Univariate and Multivariable analysis of the predictive factors of EAOC in the pre-menopausal cohort.

		Univariate Analysis		Multivariate Analysis			
		Risk Ratio (95% CI)	*p*-Value	Risk Ratio (95% CI)	*p*-Value	Risk Ratio (95% CI)	*p*-Value
CPH-I	≤6.564	1.00 (referent)		1.00 (referent)		—	—
(%)	>6.564	12.00 (3.95–36.45)	<0.001	6.45 (1.47–28.22)	0.013	—	—
ROMA Index	≤24.78	1.00 (referent)				—	—
(%)	>24.78	54.00 (6.37–457.62)	<0.001			—	—
R2 Predictive	≤16.95	1.00 (referent)		1.00 (referent)		—	—
Index	>16.95	42.50 (12.29–146.95)	<0.001	31.19 (8.48–114.74)	<0.001	—	—
Gravida	0	1.00 (referent)					
	≥1	1.04 (0.41–2.59)	0.929				
Parity	0	1.00 (referent)					
	≥1	1.25 (0.50–3.14)	0.627				
Laterality	Uni-	1.00 (referent)				1.00 (referent)	
	Bi-	0.19 (0.05–0.71)	0.013			0.15 (0.02–0.81)	0.028
BMI	≤21.94	1.00 (referent)					
	>21.94	3.55 (1.38–9.10)	0.008				
Tumor diameter	<97.50	1.00 (referent)		—	—	1.00 (referent)	
(mm)	≥97.50	14.18 (4.65–43.17)	<0.001	—	—	11.78 (3.09–44.93)	<0.001
D-dimer	<0.65	1.00 (referent)					
(µg/mL)	≥0.65	6.09 (1.93–19.26)	0.002				
CEA	<1.55	1.00 (referent)		—	—		
(ng/mL)	≥1.55	4.80 (1.73–13.25)	0.002	—	—		
HE4	<82.90	1.00 (referent)		—	—	1.00 (referent)	
(pmol/L)	≥82.90	54.00 (6.37–457.42)	<0.001	—	—	47.94 (4.01–572.03)	0.002

CPH-I Copenhagen index, ROMA risk of ovarian malignancy algorithm, BMI body mass index, CEA carcinoembryonic antigen, HE4 human epididymis protein 4.

**Table 6 diagnostics-12-01212-t006:** Accuracy analysis among the three indexes.

Index	Cohort	PLR	NLR	DOR
CPH Index	Pre-menopause	6.50	0.54	12.00
	Combined	8.41	0.41	20.16
ROMA Index	Pre-menopause	34.12	0.63	54.00
	Combined	69.12	0.28	244.28
R2 Predictive Index	Pre-menopause	11.37	0.26	42.50
	Combined	10.24	0.38	26.66

PLR positive likelihood ratio, NLR negative likelihood ratio, DOR diagnostic odds ratio, CPH-I Copenhagen index, ROMA risk of ovarian malignancy algorithm.

**Table 7 diagnostics-12-01212-t007:** Univariate and multivariable analysis of the discriminating factors of borderline tumor from OE in the combined cohort.

		Univariate Analysis		Multivariate Analysis			
		Risk Ratio (95% CI)	*p*-Value	Risk Ratio (95% CI)	*p*-Value	Risk Ratio (95% CI)	*p*-Value
CPH-I	≤6.564	1.00 (referent)				—	—
(%)	>6.564	4.23 (0.71–25.02)	0.111			—	—
ROMA Index		1.00 (referent)				—	—
(%)		57.00 (4.99–650.89)	0.001			—	—
R2 Predictive	≤16.95	1.00 (referent)		1.00 (referent)		—	—
Index	>16.95	45.00 (7.43–272.52)	<0.001	45.00 (7.43–272.52)	<0.001	—	—
Gravida	0	1.00 (referent)					
	≥1	2.23 (0.50–9.89)	0.289				
Parity	0	1.00 (referent)					
	≥1	2.65 (0.59–11.78)	0.198				
Laterality	Uni-	1.00 (referent)					
	Bi-	0.46 (0.09–2.43)	0.366				
BMI	≤21.94	1.00 (referent)					
	>21.94	5.00 (1.11–22.50)	0.036				
Tumor diameter	<97.50	1.00 (referent)		—	—	1.00 (referent)	
(mm)	≥97.50	7.62 (1.50–38.74)	0.014	—	—	7.33 (1.32–40.48)	0.022
D-dimer	<1.15	1.00 (referent)					
(µg/mL)	≥1.15	3.51 (0.75–16.38)	0.110				
CEA	<1.65	1.00 (referent)		—	—		
(ng/mL)	≥1.65	5.22 (1.00–27.31)	0.050	—	—		
HE4	<54.65	1.00 (referent)		—	—		
(pmol/L)	≥54.65	1.95 (0.35–10.66)	0.440	—	—		
CA125	<146.15	1.00 (referent)		—	—		
(U/mL)	≥146.15	3.24 (0.69–15.01)	0.133	—	—		

CPH-I Copenhagen index, ROMA risk of ovarian malignancy algorithm, BMI body mass index, CEA carcinoembryonic antigen, HE4 human epididymis protein 4, CA125 carbohydrate antigen125.

**Table 8 diagnostics-12-01212-t008:** The validation of R2 predictive index among tumor phenotypes.

	OE	Borderline Tumor	Carcinoma	*p*-Value
Number	*n* = 96	*n* = 8	*n* = 67	
R2 Predictive Index				
Median (range)	19.80 (11.47–23.32)	13.27 (−20.60–20.43)	15.16 (−12.74–25.56)	
Mean ± SD	19.61 ± 2.13	6.81 ± 16.30	14.15 ± 7.00	0.001

OE ovarian endometrioma.

## Data Availability

The data presented in this study are available on request from the corresponding author.

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
