# Peer review of "The Comparison of Three Predictive Indexes to Discriminate Malignant Ovarian Tumors from Benign Ovarian Endometrioma: The Characteristics and Efficacy"

_diagnostics, 2022, doi:10.3390/diagnostics12051212_

Round 1
Reviewer 1 Report
Dear authors,
thank you for sending your manuscript „The comparison of three predictive indexes to discriminate malignant ovarian tumors from benign ovarian endometrioma: The characteristics and efficacy“ to Diagnostics. You evaluate the prediction efficacy of malignant transformation of ovarian endometrioma using different indexes which is an important research question given the frequency of ovarian tumors and the impact on prognosis.
In the title of your manuscript you speak about the discrimination of malignant ovarian tumors from endometriomas. Did you analyze only endometriosis associated ovarian cancer or also other types of ovarian cancer in your study? Did you only consider endometriomas as benign ovarian tumors or did you include other types of benign tumors too? Reading your manuscript I do not understand this point. Summarizing my question is: did all your cancer cases develop from endometriosis and if so how do you prove this?
Perhaps you could specify your inclusion criteria please. Did all women get an ultrasound evaluation before surgery? Was the sonographic presence of endometriomas determined in accordance with the recommendations of the International Ovarian Tumor Analysis (IOTA) group? Or how did you diagnose endometriosis respectively?
Why did you choose precisely these 3 algorithms (CPH-I, ROMA and R2), as there are others too and the possibility of ultrasound?
Let me ask further questions or write comments along the text.
Introduction:
Page 1 line 36: please delete „recent“, the two molecular and morphologic categories of ovarian cancer are known since >10 years…
Page 2 lines 44-49: These informations are interesting but not the subject of the paper, I would recommend to remove
Page 2 line 49: EAOC arises from OE to a certain percentage! (10-50%) cf. Davis M et al. Comparison of clinical outcomes of patients with clear cell and endometrioid ovarian cancer associated with endometriosis to papillary serous carcinoma of the ovary. Gynecol Oncol 2014; 132: 760–766.
Page 2 line 63: please explain the R2 index in more detail
Page 2 line 65: please discuss here the option of performing ultrasound to differentiate between endometriomas and malignant ovarian tumors and cite relevant literature eg publications of the International Ovarian Tumor Analysis (IOTA) group, Moro F et al. Imaging in gynecological disease (13): clinical and ultrasound characteristics of endometrioid ovarian cancer. Ultrasound Obstet Gynecol 2018; 52: 535–543, Cohen Ben-Meir L et al. External Validation of the IOTA Classification in Women with Ovarian Masses Suspected to Be Endometrioma. J Clin MedY 2021 Jul 1;10(13):2971 and others.
Materials and methods:
Page 2 line 71: „a list of patients“ - did you include all consecutive patients being treated for ovarian tumors at your institution or did you exclude some patients? Did all patients have histologically-proven endometriosis? Also the patients of the malignant tumor group? Which stage? Did all patients of the benign tumor group have endometriomas or also other benign ovarian tumors?
Page 2 line 105: please explain the abbreviations ROMA and CPH and explain these indexes in more detail in the introduction
Results:
Page 10 line 201: table 6: a univariate and multivariate analysis in 9 cases seems difficult, this should be clearly stressed in the limitations’ section of the discussion.
Page 10 line 207: please contrast all 3 methods here and provide box plots also for ROMA and CPH.
Discussion:
Page 11 line 215: please discuss here other publications about this very subject such as: Doan Tu Tran et al. Copenhagen Index versus ROMA in preoperative ovarian malignancy risk stratification: Result from the first Vietnamese prospective cohort study. Gynecol Onco 2021;162(1):113-119. Yoshida A et al. Comparing the Copenhagen Index (CPH-I) and Risk of Ovarian Malignancy Algorithm (ROMA): Two equivalent ways to differentiate malignant from benign ovarian tumors before surgery? Gynecol Oncol 2016;140(3):481-5. Page 11 line 217: „EAOC“ or any ovarian tumor?
Page 11 line 251: „we can discriminate benign OE from malignant ovarian tumors“ - this is a mere hypothesis, you should write: „we MAY discriminate…“
Page 11 line 254: you write that the number of benign cases of post-menopausal patients is too small to assess the effectiveness of the indexes, however you do analyze it, at least for the combined cohort?
Thank you.
Best regards
Author Response
Dear reviewer,
The following are our point-by-point replies. Please check it.
In the title of your manuscript you speak about the discrimination of malignant ovarian tumors from endometriomas. Did you analyze only endometriosis associated ovarian cancer or also other types of ovarian cancer in your study? Did you only consider endometriomas as benign ovarian tumors or did you include other types of benign tumors too? Reading your manuscript I do not understand this point. Summarizing my question is: did all your cancer cases develop from endometriosis and if so how do you prove this?
-> We should explain the above information more in detail. We added them in the manuscript.
Perhaps you could specify your inclusion criteria please. Did all women get an ultrasound evaluation before surgery? Was the sonographic presence of endometriomas determined in accordance with the recommendations of the International Ovarian Tumor Analysis (IOTA) group? Or how did you diagnose endometriosis respectively?
Why did you choose precisely these 3 algorithms (CPH-I, ROMA and R2), as there are others too and the possibility of ultrasound?
-> Yes. All patients underwent ultrasound evaluation. Because physicians in Japan tend to evaluate tumors by using MRI images rather than ultrasound when newly administrated to the outpatient for surgery, the tumors did not evaluate by the IOTA system. We selected the major algorithms which use objective factors such as tumor diameter or tumor markers rather than relatively subjective findings by ultrasound.
Let me ask further questions or write comments along the text.
Introduction:
Page 1 line 36: please delete „recent“, the two molecular and morphologic categories of ovarian cancer are known since >10 years…
-> We corrected it in accordance with the reviewer.
Page 2 lines 44-49: These information are interesting but not the subject of the paper, I would recommend to remove
-> We corrected it in accordance with the reviewer.
Page 2 line 49: EAOC arises from OE to a certain percentage! (10-50%) cf. Davis M et al. Comparison of clinical outcomes of patients with clear cell and endometrioid ovarian cancer associated with endometriosis to papillary serous carcinoma of the ovary. Gynecol Oncol 2014; 132: 760–766.
-> It should be impossible to determine the tumor as arising from endometriosis in all cases by pathological examination, but it is reported that most EAOC arises from endometriosis. (Murakami K, et. al., Endometriosis-associated ovarian cancer occurs early during follow-up of endometrial cysts. Int J Clin Oncol. 2020 Jan;25(1):51-58). We changed the sentence and added the percentage of EAOC harboring the endometriosis background in the current study.
Page 2 line 63: please explain the R2 index in more detail
->We mentioned the index in more detail.
Page 2 line 65: please discuss here the option of performing ultrasound to differentiate between endometriomas and malignant ovarian tumors and cite relevant literature eg publications of the International Ovarian Tumor Analysis (IOTA) group, Moro F et al. Imaging in gynecological disease (13): clinical and ultrasound characteristics of endometrioid ovarian cancer. Ultrasound Obstet Gynecol 2018; 52: 535–543, Cohen Ben-Meir L et al. External Validation of the IOTA Classification in Women with Ovarian Masses Suspected to Be Endometrioma. J Clin MedY 2021 Jul 1;10(13):2971 and others.
->We mentioned the IOTA.
Materials and methods:
Page 2 line 71: „a list of patients“ - did you include all consecutive patients being treated for ovarian tumors at your institution or did you exclude some patients? Did all patients have histologically-proven endometriosis? Also the patients of the malignant tumor group? Which stage? Did all patients of the benign tumor group have endometriomas or also other benign ovarian tumors?
->We mentioned the above information in accordance with the reviewer's suggestions.
Page 2 line 105: please explain the abbreviations ROMA and CPH and explain these indexes in more detail in the introduction
->We added the abbreviations and explanation in the introduction.
Results:
Page 10 line 201: table 6: a univariate and multivariate analysis in 9 cases seems difficult, this should be clearly stressed in the limitations’ section of the discussion.
->We agree. We added this to the limitations.
Page 10 line 207: please contrast all 3 methods here and provide box plots also for ROMA and CPH.
-> We added the results in accordance with reviewer 1.
Discussion:
Page 11 line 215: please discuss here other publications about this very subject such as: Doan Tu Tran et al. Copenhagen Index versus ROMA in preoperative ovarian malignancy risk stratification: Result from the first Vietnamese prospective cohort study. Gynecol Onco 2021;162(1):113-119. Yoshida A et al. Comparing the Copenhagen Index (CPH-I) and Risk of Ovarian Malignancy Algorithm (ROMA): Two equivalent ways to differentiate malignant from benign ovarian tumors before surgery? Gynecol Oncol 2016;140(3):481-5.
->I agree. We corrected the sentence and added references with reviewer 1.
Page 11 line 217: „EAOC“ or any ovarian tumor?
->Yes. We corrected the sentence.
Page 11 line 251: „we can discriminate benign OE from malignant ovarian tumors“ - this is a mere hypothesis, you should write: „we MAY discriminate…“
->Yes. We corrected the sentence.
Page 11 line 254: you write that the number of benign cases of post-menopausal patients is too small to assess the effectiveness of the indexes, however you do analyze it, at least for the combined cohort?
->Yes, we showed the result in table 4.
Sincerely.
Reviewer 2 Report
The authors aimed to compare 3 predictive indexes of malignant transformation of ovarian endometrioma (OE) using the Copenhagen index (CPH-I), the risk of Ovarian Malignancy Algorithm (ROMA), and the R2 predictive index. A total of 116 patients (?) were included in this study. In this current study, cases were divided into three cohorts, pre-menopausal, post-menopausal, and combined cohort. From the data the authors concluded that: in pre-menopausal cases or borderline cases, the R2 predictive index is useful and in post- 26 menopausal cases, the ROMA index is better than other indexes.
Major Strength
- A research team dedicated to this topic with a good track record.
- Malignant transformation of ovarian endometrioma is an important issue.
- R2 predictive index is novel.
- Subgroup analysis of the pre- and post-menopausal groups.
Major Weakness
- The major weakness is the study design. Comparison of three predictive indexes should be done with diagnostic performance comparison.
- A potential bias may arise from the patient selection is biased. The R2 predictive index is proposed from the same hospital based on the patient cohort from November 2012 to February 2019. The present study is based on the patient cohort from January 2008 and July 2021. The author should declare how many cases in the present study have already been reported in the prior study to develop the R2 predictive index. (Reference: Kawahara N, Miyake R, Yamanaka S, Kobayashi H. A Novel Predictive Tool for Discriminating Endometriosis Associated Ovarian Cancer from Ovarian Endometrioma: The R2 Predictive Index. Cancers (Basel) 2021;13(15). doi: 10.3390/cancers13153829).
Specific comments:
- More detail should be given on how to measure the tumor size for the R2 predictive index in the Material and Methods.
- In the abstract: A total of 116 patients were 13 included in this study. But in the Results: a total of 172 patients included in this study were divided as follows: 116 with menstruation in the pre-menopausal cohort and 56 patients without menstruation in the post-menopausal cohort. Please clarify.
- Table 3. The cut-off values discriminating EAOC from benign OE in the pre-, post-menopausal, and combined cohort: Please explain why some data are missing. p-Value (supposed from the AUC) can be confusing here.
- Table 4 and 5 can be misleading. An additional table with comparison of three predictive indexes is strongly suggested. Please adhere to the STARD Checklist.
End of Review
Author Response
Dear reviewer,
The following are our point-by-point replies. Please check it.
- The major weakness is the study design. Comparison of three predictive indexes should be done with diagnostic performance comparison.
- A potential bias may arise from the patient selection is biased. The R2 predictive index is proposed from the same hospital based on the patient cohort from November 2012 to February 2019. The present study is based on the patient cohort from January 2008 and July 2021. The author should declare how many cases in the present study have already been reported in the prior study to develop the R2 predictive index. (Reference: Kawahara N, Miyake R, Yamanaka S, Kobayashi H. A Novel Predictive Tool for Discriminating Endometriosis Associated Ovarian Cancer from Ovarian Endometrioma: The R2 Predictive Index. Cancers (Basel) 2021;13(15). doi: 10.3390/cancers13153829).
-> We declared the shared case number with the previous study.
Specific comments:
- More detail should be given on how to measure the tumor size for the R2 predictive index in the Material and Methods.
-> We added the sentence about how to measure tumor size.
- In the abstract: A total of 116 patients were 13 included in this study. But in the Results: a total of 172 patients included in this study were divided as follows: 116 with menstruation in the pre-menopausal cohort and 56 patients without menstruation in the post-menopausal cohort. Please clarify.
-> It was a mistake. We corrected.
- Table 3. The cut-off values discriminating EAOC from benign OE in the pre-, post-menopausal, and combined cohort: Please explain why some data are missing. p-Value (supposed from the AUC) can be confusing here.
-> We analyzed the only factors which had significant differentiation for AUC.
- Table 4 and 5 can be misleading. An additional table with comparison of three predictive indexes is strongly suggested. Please adhere to the STARD Checklist.
-> We agree. We inserted the additional table concerning the accuracy of these indexes.
Round 2
Reviewer 2 Report
Thanks the authors for addressing all the points we raised.